# Accelerated Sparse Neural Training: A Provable and Efficient Method to Find N:M Transposable Masks

**Itay Hubara** [†○∗]  **Brian Chmiel** [†○∗]  **Moshe Island** [†]
**Ron Banner** [†]    **Joseph (Seffi) Naor** [◇]    **Daniel Soudry** [○]

[†]Habana Labs – An Intel company, Caesarea, Israel
[○]Electrical and Computer Engineering Department - Technion, Haifa, Israel
[◇] Computer Science Department - Technion, Haifa, Israel

{ihubara, bchmiel, misland, rbanner}@habana.ai
{naor}@cs.technion.ac.il
{daniel.soudry}@gmail.com

## Abstract

Unstructured pruning reduces the memory footprint in deep neural networks (DNNs). Recently, researchers proposed different types of structural pruning intending to reduce also the computation complexity. In this work, we first suggest a new measure called mask-diversity which correlates with the expected accuracy of the different types of structural pruning. We focus on the recently suggested $N : M$ fine-grained block sparsity mask, in which for each block of $M$ weights, we have at least $N$ zeros. While $N : M$ fine-grained block sparsity allows acceleration in actual modern hardware, it can be used only to accelerate the inference phase. In order to allow for similar accelerations in the training phase, we suggest a novel transposable fine-grained sparsity mask, where the same mask can be used for both forward and backward passes. Our transposable mask guarantees that both the weight matrix and its transpose follow the same sparsity pattern; thus, the matrix multiplication required for passing the error backward can also be accelerated. We formulate the problem of finding the optimal transposable-mask as a minimum-cost flow problem. Additionally, to speed up the minimum-cost flow computation, we also introduce a fast linear-time approximation that can be used when the masks dynamically change during training. Our experiments suggest a 2x speed-up in the matrix multiplications with no accuracy degradation over vision and language models. Finally, to solve the problem of switching between different structure constraints, we suggest a method to convert a pre-trained model with unstructured sparsity to an $N : M$ fine-grained block sparsity model with little to no training. A reference implementation can be found at `https://github.com/papers-submission/structured_transposable_masks`.

## 1 Introduction

Deep neural networks (DNNs) have established themselves as the first-choice tool for a wide range of applications, including computer vision and natural language processing. However, their impressive performance comes at a price of extensive infrastructure costs — as state-of-the-art DNNs may contain trillions of parameters [10] and require thousands of petaflops [6] for the training process. For this reason, compression of DNNs training and inference process is a research topic of paramount

---

[∗]Equal contribution.

35th Conference on Neural Information Processing Systems (NeurIPS 2021).

importance in both academia and industry. The main techniques of compression include quantization [2, 34], knowledge distillation [17], and pruning [15, 23].

Pruning DNNs is one of the most popular and widely studied methods to improve DNN resource efficiency. The different pruning methods can be categorized into two different groups: unstructured and structured pruning. While the former can achieve a very high compression ratio, it usually fails in reducing the computational footprint in modern hardware. In contrast, structured pruning methods, such as block [41] or filter [23] pruning, are more hardware friendly. Unfortunately, these methods usually fail to keep the original accuracy for high compression ratios [37]. Finding an optimal structured sparsity pattern is still an ongoing research topic.

Recently, Nvidia [36] announced the A100 GPU, containing sparse tensor cores which are able to accelerate fine-grained sparse matrix multiplication. The sparse tensor cores in A100 enable a 2x acceleration of regular matrix multiplication in DNNs, $Y = WX$, where $W$ and $X$ are weight and input matrices, respectively. The only requirement is that $W$ would have a fine-grained 2:4 sparsity structure, i.e. out of every four contiguous elements in $W$, two are pruned. Nvidia [36] suggested a two-fold scheme for pruning a pretrained dense model: (a) Define a fine-grained 2:4 fixed mask, and (b) retrain with the masked weights using original training schedule. Indeed, the Nvidia [36] approach is very appealing for the common case where a pretrained dense model is given.

While the Nvidia [36] method works well on many models, a pretrained model is not always given. In those cases, one has to first train a dense model and only then try to prune it. To alleviate this demand, Zhou et al. [43] suggested a method that trains from scratch a model with $N : M$ fine-grained mask, using a sparse-refined straight-through estimator (SR-STE). Similarly to the quantization-aware-training methods [18], they maintain a dense copy of the weights and prune it in every iteration, passing the gradients using the straight-through estimator [5]. Since the mask dynamically changes while training, they suggest adding an extra weight decay on the masked (i.e. pruned) elements to reduce the mask changes during the training process. As opposed to Evci et al. [9] that aims to reduce memory footprint for sparse training from scratch, Zhou et al. [43] only eliminates the need to train a dense model before pruning it.

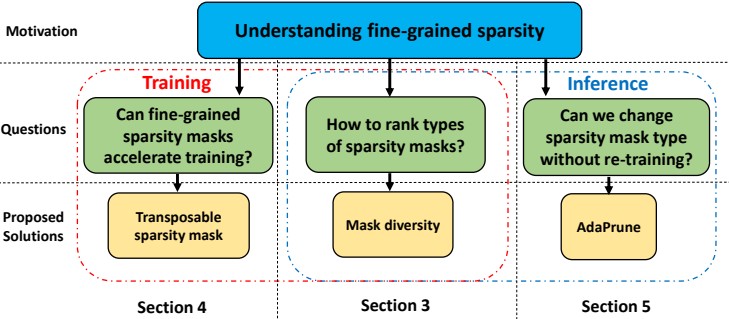

Figure 1: High-level overview of the different questions and their corresponding solutions proposed in this work. Motivated by understanding fine-grained sparsity we first suggest a measure to rank different sparsity mask, then we suggest a method to accelerate training with fine-grained sparsity and finally propose a method to change the fine-grained mask without re-training.

Motivated by these promising results, our goal here is to answer three remaining questions: (Fig. 1)

1. **How to rank different types of sparsity masks?** We suggest a new measure called "mask diversity", which is the first to connect mask constraints and network accuracy (Section 3).

2. **Can fine-grained sparsity masks accelerate training?** We start by observing both the forward and the backward matrix-multiplications involving the weight matrix $W$. Since the backward pass requires using the transposed matrix $W^T$:

$$\frac{\partial \text{Loss}}{\partial X} = W^T \cdot \frac{\partial \text{Loss}}{\partial Y}, \tag{1}$$

and in general $W^T$ does not have an $N : M$ fine-grained sparsity structure (even if $W$ has this structure), the methods suggested in [43, 36] accelerate only the forward pass matrix-multiplication,

$Y = WX$. Consequently, the current methods only utilize in part the sparse tensor cores to accelerate training. We propose a novel $N : M$ transposable-fine-grained sparsity mask, where the same mask can be used for both forward and backward passes (Fig. 2). We focus on accelerating sparse training in the two settings detailed above: (a) starting from a pretrained model, and (b) starting from scratch. For (a) we derive a novel algorithm to determine the optimal transposable-mask using a reduction to a min-cost flow problem. For (b) we devise an approximation algorithm with an (almost) linear (in input-size) time complexity that produces a mask whose $\ell_1$ norm is within a factor of 2 from the optimal mask. We show the effectiveness of both methods (Section 4).

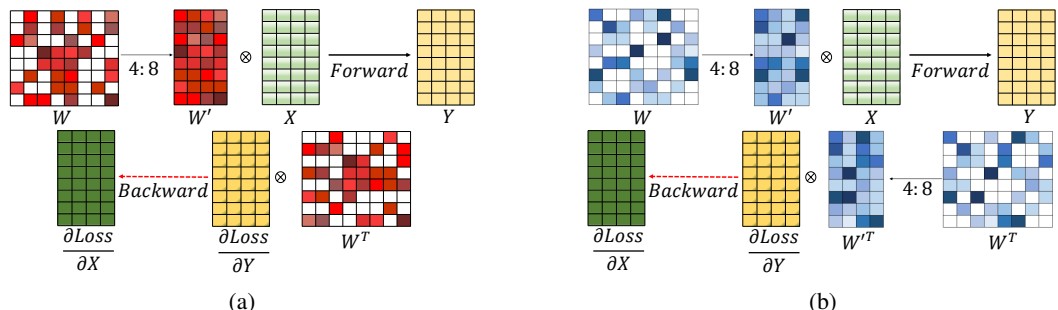

(a)                 (b)

Figure 2: **(a):** A 4:8 structured pruning mask, as in Zhou et al. [43], Nvidia [36], capable of accelerating with sparse tensors core only the forward pass. **(b):** The suggested 4:8 transposable structured pruning mask capable of accelerating with sparse tensors core the forward and backward passes.

3. **Can we change the type of sparsity mask structure without re-training?** Different hardware devices can support different types of fine-grained sparsity masks. Therefore, we suggest the "Adaprune" method, which converts between types of sparsity masks (even from unstructured masks) without the need of re-training, and almost no degradation in accuracy (Section 5).

## 2   Related work

Pruning of neural networks weights has been extensively investigated, starting with classical methods in the late 1980s [20, 31, 32, 21] and then amounting to dozens of papers published in recent years. Since DNNs are generally over-parameterized, pruning the weights reduces their memory footprint. In special cases, when the sparsity mask has a specific pattern, it has the potential to reduce computational footprint as well. The most common practice is to prune a pretrained dense model, so that it will be sparse at deployment. Since the accuracy of the pretrained dense model is known, one can tune the sparsity level to ensure comparable accuracy for the sparse model. Recently, a new line of research that aims to train sparse models from scratch [14, 7, 9] has emerged. The goal is to train models that cannot fit into currently available hardware. Next, we briefly overview the structured, unstructured, and accelerating-sparse-training categories.

**Unstructured pruning.** It removes individual elements of the matrix, aiming for high total sparsity, while being agnostic to the location of the pruned elements. Standard pruning methods are based on different criteria, such as magnitude [15], approximate $L_0$ regularization [26], or connection sensitivity [22]. Recent methods [12], suggested to train a dense network until convergence, extract the required mask ("winning ticket"), and use the original training regime to re-train the active weights from their original initialization or final values [38] using the original training schedule. These methods are able to achieve over 80% sparsity on ResNet50- ImageNet dataset [38]. Despite the high sparsity ratio that can be achieved with these methods, modern hardware cannot efficiently utilize such a form of sparsity for reducing computational resources [36].

**Structured pruning.** It removes weights in specific location based patterns, which are more useful for hardware acceleration. Such methods can be applied at either the level of channels or layers. For example, Li et al. [23] remove the channels with the lower norm, Luo et al. [27] prune channels according to the effect on the activation of the following layer, and [41] split the filters into multiple groups, applying a group Lasso regularization. All these methods are natively supported in both hardware and software, as they effectively change the model structure by reducing channels or groups. Yet, no such method was able to achieve a reasonable accuracy with sparsity levels higher than 50%. As observed by Liu et al. [25], filter pruning of a pretrained dense over-parameterized model is rarely the best method to obtain an efficient final model. Thus, here the structured pruning serves mostly

as a DNN architecture search for the optimal compact model [40, 42]. Our work is most closely related to Zhou et al. [43], which is the first work that attempted training with a fine-grained $N : M$ structured sparsity mask, as explained above.

**Accelerating sparse training for model deployment.** The most common approach for sparse model deployment requires a three steps process: (a) train a dense model; (b) define a sparsity mask (c) fine-tune while enforcing the mask on the model's weights. The lottery ticket hypothesis of Frankle & Carbin [12] demonstrates that we can avoid the dense training, i.e., step (a), had we known how to choose the appropriate mask. Since Frankle & Carbin [12] discovered the optimal mask (winning ticket) by applying dense training, the question of how to find the optimal mask without training remained open. Since setting a predefined fixed mask results in some accuracy degradation (Gray et al. [14] that requires expanding the model size), several researchers [4, 29, 30, 7, 9] tried to enable dynamic mask changes during the training process. All methods focused on unstructured sparsity and aimed to enable training large models on hardware with memory limitations. In contrast to these approaches, Zhou et al. [43] does not aim to reduce the model memory footprint during training, but rather accelerate inference while avoiding dense pre-training. To that end, they keep a dense weight matrix, and in each iteration they re-calculate the mask. With the obtained pruned copy of the weights they perform the forward and backward pass and update the dense copy. Therefore, this method is most relevant when a pretrained dense model is not given, and one wishes to obtain a fine grained $N : M$ sparse model for inference. While our work focuses on the setting of Zhou et al. [43], we argue that it can easily combined with Evci et al. [9] work, as it aims to solve a different issue within the same problem.

## 3 Mask Diversity

Structured sparsity requires masks with a hardware-friendly structure type. Yet, which structure should be employed is still an open question. The additional hardware cost (mainly chip-area and power) required to support each of the structures is hard to quantify, as it varies based on the hardware design. However, the effect of the structure on the model accuracy is oblivious to the hardware at hand. Therefore, in this section, we aim to find a method for ranking different types of sparsity masks, which can predict help the model accuracy. We start from an hypothesis that the structure constraints cause accuracy degradation. Thus, we expect that the best sparsity levels, without accuracy degradation, would be achieved by unstructured sparsity, which has no requirements on the sparsity structure type. To quantify how much a specific structure constrains the model, we introduce a new measure, called *mask-diversity* (MD). MD is the number of all possible masks that adhere to the restrictions of the structure under similar sparsity level. As an example, we derive the MD for a tensor size $T$ under four different structure constraints:

1. **Unstructured sparsity:** no constraints, except for an overall sparsity level. This is the most common setting in the literature.

2. **Structured N:M sparsity:** $N$ values in each block of size $M$ are set to zero. This structure is currently supported in the most widely deployed hardware [36].

3. **Transposable N:M sparsity:** for a block of size $M \times M$ both columns and rows must follow the $N : M$ fine-grained constraints. As we later discuss (Section 4), the transposable structure is essential for training acceleration.

4. **Sequential N:M sparsity:** any block of size $M$ must contain $N$ sequential zeros. This is an example for a small potential modification to the $N : M$ fine-grained structure, which might be more hardware friendly (data can be compressed and decompressed more easily).

MD depends on the required sparsity level (and the tensor size), thus without loss of generality we set the sparsity level to be $N/M$. In Eq. (2) we write the MD for each of the constraints (1-4) above, derived using basic combinatorial arguments (Appendix A.6):

$$
\textbf{1. } \text{MD}_{\text{Unstructured}} = \binom{T}{\frac{NT}{M}}; \qquad \textbf{3. } \text{MD}_{\text{Transposable}} = (M!\,(M-1)!\cdots(M-N+1)!)^{\frac{T}{M^2}}
$$

$$
\textbf{2. } \text{MD}_{\text{Structured}} = \left(\frac{M!}{N!\,(M-N)!}\right)^{\frac{T}{M}}; \qquad \textbf{4. } \text{MD}_{\text{Sequential}} = (M-N+1)^{\frac{T}{M}}
\tag{2}
$$

Table 1 shows the MD for a matrix of size $8 \times 8$ with 50% sparsity under different constraints. As expected, unstructured sparsity achieves best in class MD, however its hardware cost makes it unfeasible to use on modern architectures [14, 8, 13]. Additionally, sequential N:M sparsity has worst in class MD and con-

Table 1: MD for different constraints for a matrix of size $8 \times 8$.

| $N : M$ | 1:2 | 2:4 | 4:8 |
|---|---|---|---|
| Unstructured | $1.8 \cdot 10^{18}$ | $1.8 \cdot 10^{18}$ | $1.8 \cdot 10^{18}$ |
| Structured | $4 \cdot 10^{9}$ | $2.8 \cdot 10^{12}$ | $5.7 \cdot 10^{14}$ |
| Transposable | $6 \cdot 10^{4}$ | $4 \cdot 10^{8}$ | $1.7 \cdot 10^{13}$ |
| Sequential | $4 \cdot 10^{9}$ | $4 \cdot 10^{7}$ | $4 \cdot 10^{5}$ |

sequently we expect that using it would harm the model accuracy (Fig. 3a). Thus, from here on, we focus on the structured and transposable structured fine-grained sparsity. Notice the MD of structured 2:4, which was used in [36, 43] is severely reduced when transposable constraints are enforced. Since a transposable structure enables training acceleration (Section 4), supporting this structure is essential. Thus, we suggest a minimal expansion of the block size (from 4 to 8) which enables the transposable fine-grained structure to reach a similar MD as the 2:4. Increasing the block size requires more multiplexers to support the $N : M$ structure [24]. However, the hardware cost increases only logarithmically with respect to $N$ (details given in Appendix A.5). We therefore argue that such an expansion is reasonable. To better test our hypothesis, we measured the accuracy degradation with respect to the mask diversity on ResNet18 with Cifar-100 dataset. As can be seen in Fig. 3a, the accuracy correlates with the MD metric and the 4:8 transposable fine-grained mask reached a comparable accuracy to the 2:4 fine-grained mask. Moreover, recall the common method of magnitude pruning has an underlying objective to preserve the tensor $\ell_1$ norm. We therefore measured the effect of the different constraints on the $\ell_1$ norm in Fig. 3b. This figure shows the $\ell_1$ norm of the last layer in trained ResNet-50 after applying different structured pruning. Here as well, 2:4 structured and 4:8 transposable structured have (almost) similar $\ell_1$ norms.

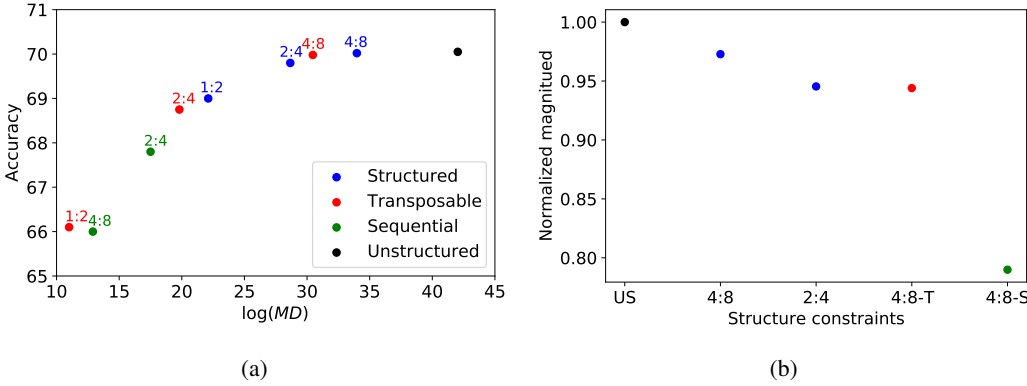

(a)                                                                 (b)

Figure 3: **(a):** ResNet18 over Cifar100 top-1 accuracy for weight sparsity of 50 % using different structured and unstructured masks. As expected the mask diversity correlates with the pruned model accuracy. **(b):** Magnitude of the last layer's weight tensor of ResNet-50 (pretrained dense model) masked with structured mask 4:8, 2:4, 4:8 transposable ("4:8-T") and 4:8 sequential ("4:8-S") normalized by the unstructured 50% sparsity ("US"). Notice that mask diversity is correlated with magnitude preservation. As expected the 4:8 transposable mask has a similar $\ell_1$ norm score as the 2:4 mask. Additional results and details in Appendix A.7 .

## 4    Computing transposable sparsity masks

In general, training DNNs requires three matrix multiplications per layer. The first multiplication is required for the forward propagation between the weights and activation. The other two multiplications are used for the backward and update phases. The backward phase calculates the gradients of the loss function with respect to the input of the neural layer. This is done by recursively passing the error from the last layer to the first (Eq. (1)). Note that the backward phase uses the transposed weight matrix. Hence, accelerating the backward phase requires the transposed weight matrix to adhere to the hardware required pattern (e.g., $N : M$ fine-grained sparsity). In this section, we tackle this issue by presenting a novel to find $N : M$ transposable fine-grained sparsity masks, where the same mask can be used to accelerate both forward and backward passes (Fig. 2). The required mask contains only $M - N$ non-zero elements, for every contiguous $M$ elements, in both $W$ and $W^T$ simultaneously. We formulate the problem and suggest two methods to generate the transposable mask.

**Problem formulation.** First, we provide an integer-programming (IP) formulation for finding an optimal transposable fine-grained mask. Let us consider a block of size $M \times M$ in a weight matrix $W$. Our goal is to maximize the $\ell_1$ norm of $W$ after masking N elements in each row and column. We formulate the problem as an integer program. Define a binary sparsity mask $S \in \{0,1\}^{M \times M}$, where $S_{i,j} = 1$ if and only if the element $W_{i,j}$ is not pruned, and otherwise $S_{i,j} = 0$. The resulting integer program is

$$\max_{S \in \{0,1\}^{M \times M}} \sum_{i,j} S_{i,j} |W_{i,j}| \ \text{ s.t. } \forall j : \sum_i S_{i,j} = N , \forall i : \sum_j S_{i,j} = N . \tag{3}$$

In the following, we examine several methods for solving this problem. We first describe an optimal, yet computationally expensive method. We then describe a more efficient method, which provides an approximate near-optimal solution.

**Reduction to Min-Cost Flow.** General integer programs (IP) require exponential time complexity with respect to the input size (worst-case). Fortunately, Fig. 4 shows that our IP formulation (Eq. (3)) reduces to a min-cost flow problem. There is a vast literature on efficient min-cost flow algorithms (see e.g., [1]), however, the most efficient algorithms for our setting take time $O(M^3 \log M)$ time for computing an optimal transposable mask for a block size of $M \times M$ [1][pp. 396-397][2]. The min-cost flow solution should be used when training from a pretrained dense model, where the transposable mask is generated once, remaining fixed from then on during training. On the other hand, sparse training from scratch requires changing the mask during training, and it is therefore essential to find a very efficient algorithm for computing the mask. To this end, we design a light *2-approximation* algorithm, i.e., for every input it produces a solution which is guaranteed to be within a factor of 2 of an optimal solution (to the given input), yet it runs in almost linear time.

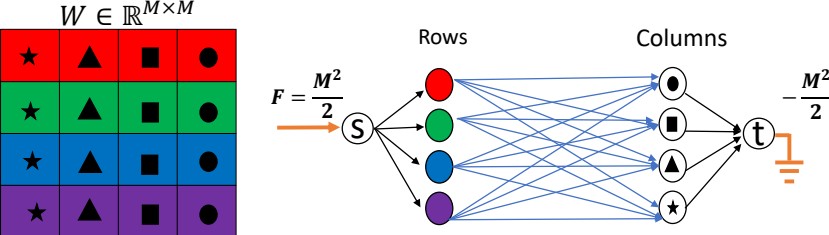

Figure 4: $\frac{M}{2} : M$ transposable-sparsity optimization as a min-cost flow problem. In addition to a source and a sink, the network has a node for each row and for each column. The construction uses three types of edges: (i) *source edges* emanating from the source node $s$ into each row node $i$; (ii) *sink edges* connecting each column node $j$ with the sink node $t$; and (iii) a *coefficient edge* $(i, j)$ for each matrix element $W_{i,j}$. Each source edge $(s, i)$ has capacity $\frac{M}{2}$ which is equal to the number of elements that need to be selected for pruning in row $i$. Similarly, each sink edge $(j, t)$ has capacity $\frac{M}{2}$ which is equal to the number of elements pruned in column $j$. Each coefficient edge $(i, j)$ has unit capacity and cost $|W_{i,j}|$. Finally, selecting a matrix element with weight $W_{i,j}$ for pruning corresponds to a unit flow on the coefficient edge $(i, j)$. Assuming the source and sink edges have zero-cost, there is a one-to-one correspondence between a min-cost flow solution that sends a flow of value $\frac{M^2}{2}$ from the source $s$ to the destination $t$ in this construction, and an optimal transposable mask minimizing the sum of absolute values selected for pruning.

**2-Approximation algorithm.** We design a greedy *2-approximation* algorithm (see Algorithm 1) having a low time complexity that can be used in practice without compromising too much the quality of the solution produced. Unlike the optimal min cost flow solution that runs in time complexity of $O(M^3)$ for a block size of $M \times M$, Algorithm 1 has a running time of $O(M^2 \log M)$ i.e., a time complexity that is almost linear in the number of block elements $M^2$. The approximation algorithm uses the same construction described in Fig. 4, but instead of running a min-cost flow on the graph, it employs a simple greedy approach. Let $P$ be the list of edges pruned by Algorithm 1, let $W(P)$ be the total weight of the edges in $P$, and let $W^*$ be the weight of an optimal solution (i.e., the minimal sum of edges that can be pruned to create a $\frac{M}{2} : M$ transposable sparsity mask). The next lemma establishes that Algorithm 1 finds a 2-approximate solution (proof in Appendix A.2, with an example showing the upper bound is tight):

---

[2]More modern methods that are based on interior point algorithms seem to be less efficient for our setting.

**Lemma.** *Algorithm 1 produces a tight 2-approximate solution, i.e., $W(P) < 2 \cdot W^*$.*

---

**Algorithm 1** 2-approximation algorithm. **Input:** Bipartite graph G=(V,E) ; **Initialize:** P= $\emptyset$

---

1: Sort list of coefficient edges from light to heavy; Let $A = [e_1, ..., e_n]$ be the sorted list of edges.
2: **for** each edge $e_i = (u, v) \in A$, $i = 1, \ldots, n$ **do**
3:      **if** degree(u)$\leq \frac{M}{2}$ or degree(v)$\leq \frac{M}{2}$ in P **then** P $\leftarrow$ P + $e_i$.

---

In Table 2 we show the running time overhead of ResNet50 training with IP, min cost flow and 2-approximation algorithms over regular training, the algorithms were implemented in a non-optimized way. All experiments were run in a single GPU and the mask was updated every 40 iterations. Notice the acceleration achieved with the 2-approximation algorithm in comparison to the naive IP. In Appendix A.3 we extend the complexity analysis.

## 4.1 Experiments

In this section, we demonstrate the effectiveness of our proposed transposable $N : M$ fine-grained structured sparsity in computer vision and natural language processing tasks. We evaluate the suggested method in two cases: (i) initialize from a trained dense model and re-train with a fixed mask, similar to APEX's Automatic Sparsity (ASP [36]), (ii) train from scratch and update the mask frequently, as done by Zhou et al. [43]. We show comparable accuracy to previous methods, while

Table 2: Overhead of different algorithms for finding the $4 : 8$ transposable mask: ratio of their running time over regular training (ResNet50). Notice the overhead reduction in the 2-approximation algorithm in comparison to the naive IP.

| Method | Overhead (%) |
|---|---|
| Integer-programming | 180 |
| Min-cost flow | 70 |
| 2-approximation | 14 |

achieving a significant reduction of the training process resources — by exploiting the sparse tensor core abilities, allowing their use both in forward and backward passes. In all the experiments we use a 4:8 transposable-mask, which as shown in Table 1, have a similar MD as 2:4 mask used in previous works [36, 43]. Experimental details appear in Appendix A.4. In Appendix A.8 we show additional experiments with the 2:4 transpose mask showing that: (1) in most cases, 2:4 transpose is enough to achieve high accuracy, (2) in some cases where the 4:8 transpose is necessary, and (3) this is consistent with the MD results shown in Section 3.

**Initialization from a trained dense model** We evaluate the suggested $N : M$ transposable mask using a trained dense model as initialization. In order to find the transposable mask, we solve the min-cost flow reduction (Fig. 4) on the dense trained network and then fix the mask. In Table 3 we compare our method with ASP [36] on classification (ResNet50 - ImageNet dataset), detection (MaskRCNN - COCO dataset) and question answering (BERT-large - SQuAD dataset) tasks. While both methods initialized from the pre-trained models, the 4:8 transposable enables propagation acceleration in the retraining phase where ASP does not.

Table 3: Comparison of our suggested method with ASP [36] initialized from dense model on ResNet50 (Imagenet dataset), BERT-large (SQuAD dataset) and MaskRCNN (COCO dataset). We use a transposable 4:8 mask while ASP used 2:4. SU refers to the sparse tensor cores utilization, used both in forward and backward passes in the proposed method, allowing 2x sparse tensor cores utilization in comparison to ASP which use the sparse tensor cores only in the forward pass. Missing results in Nvidia [36] were marked by '-'

| Model | Metric | Baseline | | ASP[36] 2:4 | | Ours 4:8-T | |
|---|---|---|---|---|---|---|---|
| | | SU | Accuracy | SU | Accuracy | SU | Accuracy |
| ResNet18 | Top1 | | 69.7% | | - | | 70.06% |
| ResNet50 | Top1 | 0% | 76.15% | 33% | 76.6% | 66% | 76.6% |
| Bert-Large | F1 | | 91.1 | | 91.5 | | 91.67 |
| MaskRCNN | AP | | 37.7 | | 37.9 | | 37.84 |

**Sparse Training from scratch.** In order to avoid the training of a dense model, we also evaluate the proposed transposable $N : M$ mask in the training from scratch setting. Similar to Zhou et al. [43] we keep a dense copy of the weights and before each forward pass we mask the weights with a $N : M$ transposable mask. In contrast to Zhou et al. [43] who changed the mask every iteration, we

Table 4: Training from scratch of ResNet18, ResNet50, ResNext50 and Vgg11 on imageNet dataset and fine-tuning of Bert-base on SQuAD dataset, using the proposed 2-approximation scheme. SU refers to the sparse tensor cores utilization, used both in forward and backward passes in the proposed method, allowing 2x sparse tensor cores utilization in comparison to N:M-SS [43] with comparable results. Missing results in Zhou et al. [43] were marked by '-'

| Model | Metric | Baseline | | 4:8-SS [43] | | Ours 4:8-T | |
|---|---|---|---|---|---|---|---|
| | | SU | Accuracy | SU | Accuracy | SU | Accuracy |
| ResNet18 | Top1 | | 70.54% | | 71.2% | | 70.75% |
| ResNet50 | Top1 | | 77.3% | | 77.4% | | 77.1% |
| ResNext50 | Top1 | 0% | 77.6% | 33% | - | 66% | 77.4% |
| Vgg11 | Top1 | | 69% | | - | | 68.8% |
| BERT-base | F1 | | 88.52 | | - | | 88.38 |

found that we can use 2-approximation scheme to extract the transposable mask every 40 iterations. Empirically we found that the 2-approximation scheme is on average within a factor of 1.2 from the optimal mask. The hyper-parameters used for training are equal to the ones suggested by Zhou et al. [43]. In Table 4 we test the proposed method over ResNet18, ResNet50, ResNext50, Vgg11 (ImageNet dataset) and fine-tune of Bert (SQuAD-v1.1 dataset) and compare to Zhou et al. [43] results. As can be seen, we achieved comparable accuracy with 2x sparse tensor cores utilization in the training process.

## 5 Structured sparsity without full training

In section Section 4 we suggested 4:8 transposable-mask to accelerate training, but what should we do if we wish to deploy the output of such training on hardware that supports only 2:4 fine-grained sparsity. Forcing structured sparsity on a model that was trained with a different structured sparsity, leads to a severe accuracy degradation as several bits of the mask may change to satisfy the structured sparsity requirements. In this section we would focus on the more common case of deploying unstructured sparse model on hardware that support N:M structure. This is a fundamental problem as most DNN pruning methods focus on unstructured pruning, which reduces the memory footprint. However, current hardware implementations suggest that, unless very high sparsity levels are achieved, the model cannot be accelerated at all. Hence, commonly, the weights are simply decompressed before multiplication. To understand the problem we study the probability that an unstructured mask would not violate any $N : M$ constraint [36]. Then we discuss two light methods to bridge the gap when a sparse model is given but the hardware does not support its structure.

**Probability for violating the $N : M$ constraint in unstructured sparsity.** Let $X = \{x_1, x_2, ..., x_M\}$ be a block of independent and identically distributed random variables. Assume that with a probability $\rho$, $x_i$ can be pruned without accuracy degradation (i.e., unstructured pruning). In this section, we consider a general form of block sparsity in which, for a block of size $M$, at least $N$ values could be pruned. Define $X$ to be $N : M$ sparse if this $M$ sized block has at least $N$ values that can be pruned without harming accuracy. The probability of having a $N : M$ sparse block is given by the binomial distribution and so

$$P\left(X \text{ is } N : M \text{ sparse}\right) = \sum_{i \geq N} \binom{M}{i} \cdot \rho^i \cdot (1 - \rho)^{M-i} \tag{4}$$

In Fig. 5a we plot Eq. (4) for the case of $\rho = 0.5$. To force a given sparse model to have a fine grained $N : M$ sparsity, we need to make sure that $N$ out of every $M$ contiguous elements are zero. Therefore, as in Nvidia [36], in each block we prune $N$ weights with the lowest magnitude (including any zero weights, e.g., non-active). Forcing this pattern on an existing unstructured mask might remove active (non-zero) weights, i.e., flipping some of the mask values from one to zero. We named those required flips, *pattern-violations*. Removing active weights without re-training tends to severely degrade the model accuracy. To demonstrate the problem we used an unstructured sparse pretrained ResNet-50 model ($\rho = 0.86$) and set the $N : M$ structure per-layer, based on Eq. (4), such that the probability for a pattern-violation would be equal or less than a given percentage. Here we used a block size of $M = 8$. Notably, without any optimization even a $1\%$ pattern-violation results in severe degradation (Fig. 5b). Next, we describe two light methods to boost the accuracy.

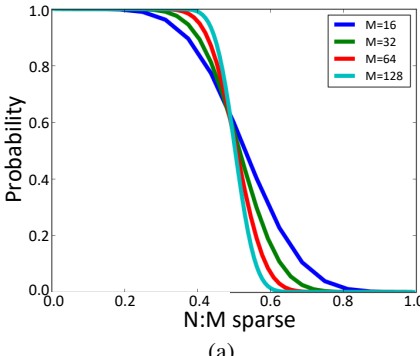
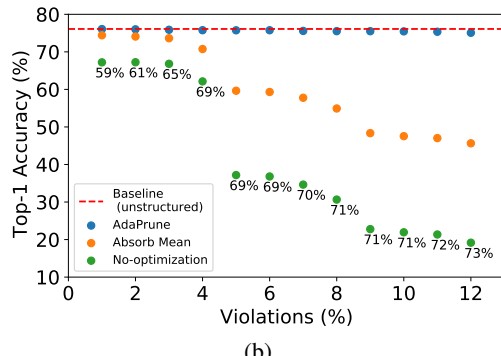

|        |        |
| (a)    | (b)    |

Figure 5: **(a):** Eq. (4) for $\rho = 0.5$ and various block sizes $M$. We have a sharp ("phase") transition at $N/M = \rho$. Specifically, (i) when $N/M \leq \rho$ we have a probability larger than 0.5 that the sampled block is $N : M$ sparse; (ii) when $N/M \geq \rho$ this probability quickly decreases to zero. As block size $M$ increases this phase-transition gets sharper. As expected, when $M \to \infty$, unstructured sparsity satisfies the structured constraints, and we expect it to display the phase transition precisely at the critical point $\rho$. **(b):** Top-1 accuracy vs. percent of constraints violated. The numbers next to the baseline samples represents the sparsity level of the refined model.

**Fixing pruning bias using mean absorption.**    Several works [3, 11, 19] reported that it is important to fix the bias introduced when quantizing the model. We build on those results and suggest absorbing the mean of the $N$ pruned weights into the $M - N$ non zeroed weights. As can be seen in Fig. 5b this simple fix, by itself, greatly boosts accuracy.

**AdaPrune.**    Recently, several works [19, 33] suggested light and fast fine-tuning techniques for post-train quantization. These techniques replace the heavy full model training with a fast per-layer optimization which requires only a few iterations to converge. While each method applies a different optimization technique, they all aim to reduce the discrepancy between the quantized and full-precision layer outputs. We adjusted the parallel-AdaQuant [19] technique to the pruning problem by defining its objective to be:

$$\min_{W'} ||WX - (S \odot W')X||_2^2, \tag{5}$$

where $W$ is the original weight layer, $W'$ is the weight layer we aim to find, $X$ is the output of the previous activation layer, $S$ is the weight sparsity mask, and $\odot$ is a component-wise product. We named this method AdaPrune. In our experiments we used 1000 images from the ImageNet training set as a calibration set. As can be seen in Fig. 5b, AdaPrune is capable of correcting the remaining error and obtain less than 1% degradation from the original unstructured-sparse model counterpart. We argue that with AdaPrune, we can potentially adapt any generic mask to the hardware at hand, thus elevate the need to retrain the model. However, full re-training is still necessary when starting from a dense model (thus having 50% pattern-violation), since there we get 2.3% degradation using AdaPrune. We discuss and extend those experiments in Appendix A.1.

## 6    Conclusions

In this work, we analyze the constraints introduced by fine-grained sparsity. We discuss the limitations of current research in the field and suggest a new measure called *mask diversity* to connect between the mask constraint and the accuracy of the network. In addition, we discuss the inherent problem of accelerating sparse training and suggest a novel N:M transposable-mask which enables accelerating the backward phase as well. We formulate the question of finding the optimal mask as a minimum-cost-flow problem and show no accuracy degradation in a variety of tasks with 2x acceleration in comparison to previous methods [36]. We experimented with transposable-masks also in the setting of Zhou et al. [43] where pretrained model is not given but the mask is dynamically changing based on the gradients and a copy of the weights which are kept dense. In this setting we define an approximation scheme with an (almost) linear (in input-size) time complexity that produces a mask whose $\ell_1$-norm is within a factor of 2 from the optimal mask and allow 2x acceleration in comparison to Zhou et al. [43]. While a different line of research suggested more extreme setting which restrict the memory footprint to the compressed model size [4, 29, 30, 7, 9], we argue that our method is orthogonal to it and both methods could be easily be combined. We believe this work paves the path toward true efficient sparse training. Finally, we suggest a simple method (AdaPrune) to transform an

unstructured sparse model to an $N : M$ fine-grained sparse structured model with little to no training (e.g., less than 1% degradation in ResNet50). Furthermore, with a light training procedure over a calibration set (i.e., AdaPrune) we can switch from unstructured model to 4:8 structure sparsity and thus gain 2x inference speedup.

**Broader impact.** Training DNNs is an expensive process which requires a high amount of resources and time. This long time can prevent the use of DNNs in many applications, despite their impressive performance. Reducing training time gives the opportunity to to use of DNNs in additional applications, or even use the extra time to improve the existing models. We need to take into account that accelerated training could introduce training instabilities. These models will need a careful examination, specially when used in real applications, such as medical devices.

## Acknowledgements

The research of JN is supported in part by US-Israel BSF grant 2018352 and by ISF grant 2233/19 (2027511). The research of DS was supported by the Israel Science Foundation (grant No. 1308/18), and by the Israel Innovation Authority (the Avatar Consortium).

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
