# A  Supplementary Material

## A.1   Additional AdaPrune experiments

To further examine AdaPrune capabilities we checked two additional settings: (a) starting from pre-trained dense model, and (b) staring from less constrained N:M mask.

### A.1.1   AdaPrune from Dense

While this case is more common, we expect to see some degradation as we know that we have 50% mask violations. Yet as can be seen in table 2 we managed to restore accuracy to 2-3% of the full-precision baseline using just AdaPrune. To further improve results we applied batch-norm-tuning as suggested by Hubara et al. [19] and kept the first and last layers dense which results in less than 2% degradation. We believe it to be the first tolerable post-training-pruning results reported.

Table A.1: Using AdaPrune from dense pre-trained model. AP stands for AdaPrune and BNT stands for batch-norm-tuning.

| Model | Dense | BiasFix BNT | AP | AP BNT |
|---|---|---|---|---|
| ResNet18 | 69.7% | 62.47% | 68.41% | 68.63% |
| ResNet34 | 73.3% | 68.72% | 72.15% | 72.36% |
| ResNet50 | 76.1% | 67.42% | 74.41 | 74.75% |
| ResNet101 | 77.27 % | 71.54% | 76.36% | 76.48% |

### A.1.2   AdaPrune from N:M sparse

In Section 3 we explained why as the block size decreases the mask diversity decreases. Thus, we expect to have many violations when a pre-trained sparse model with $N_1 : M_1$ translates to $N_2 : M_2$, for $N_1 > N_2$ and $M_1 > M_2$. We argue that this might be a common case in the future as different hardware vendors would support different formats. In table Table A.2 we can see results of converting ResNet-50 model trained with $4 : 8$ sparsity pattern to $2 : 4$ and $1 : 2$ patterns. As can be seen, converting from $4 : 8$ to $2 : 4$ produces results with negligible accuracy degradation (less than 0.5%). Therefore, we argue that AdaPrune is an efficient and useful approach to convert models which were optimized on a different hardware than the one in use, as it removes the need for full sparse training. This is even more important when the training data is not available.

Table A.2: Using AdaPrune to convert from one sparse pattern to the other. The baseline model was trained with 4:8 sparsity (90 epochs). Thus, 4:8 column is the baseline. BNT stands for batch-norm-tuning

| Model | 4:8 | 2:4 | 2:4 BNT | 1:2 | 1:2 BNT |
|---|---|---|---|---|---|
| RN50 | 76.5% | 76.2% | 76.4% | 74.6% | 75.1% |
| RN50-T | 77.1% | 76.3% | 76.4% | 74.7% | 75.1% |
| RN18-T | 70.75% | 70.1% | 70.2% | 68.9% | 69.2% |

## A.2   Proof of Lemma

**Lemma.** *Algorithm 1 produces a tight 2-approximate solution, i.e., $W(P) < 2 \cdot W^*$.*

*Proof.* Consider any node $i \in V \setminus \{s, t\}$. Let $E'(i) = \{e'_1, e'_2, e'_3, ...e'_{M/2}\}$ denote the edges of an optimal solution that are adjacent to node $i$ and sorted in ascending order from light to heavy. Let $E(i) = \{e_1, e_2, e_3, ...e_{M/2}\}$ denote the first $M/2$ edges adjacent to $i$ in $P$ with respect to the order in which Algorithm 1 picked them. By construction, we have that for all edges in $E(i)$:

$$w(e_1) \leq w(e_2)... \leq w(e_{\frac{M}{2}}). \tag{A.1}$$

We note that we can truncate the list of $i$ at $M/2$, since if $i$ has more than $M/2$ edges adjacent to it in $P$, then any such edge $(i, j)$ would also appear in $E(j)$ (among the first $M/2$ edges adjacent to $j$) by the minimality of the solution $P$. Thus, the union of the lists $E(i)$ contains all edges in $P$. We now prove by induction that for any $n$, $n \geq 1$,

$$w(e_n) \leq w(e'_n). \tag{A.2}$$

- Base case ($n = 1$): $w(e_1) \leq w(e'_1)$, since by construction of Algorithm 1, edge $e_1$ is the lightest edge adjacent to node $i$.

- Inductive step: assume $w(e_n) \leq w(e'_n)$, then it must hold that $w(e_{n+1}) \leq w(e'_{n+1})$); otherwise, if $w(e_{n+1}) > w(e'_{n+1})$), then $e'_{n+1}$ should have been considered before $e_{n+1}$ and also chosen by Algorithm 1.

Thus,

$$\sum_{j=1}^{M/2} w(e_j) \leq \sum_{j=1}^{M/2} w(e'_j).$$

To complete the proof, our goal is to charge the weight of the edges in $P$ to the weight of the edges in the optimal solution based on the above inequality. However, note that an edge $(i, j) \in P$ may appear in only one of the lists $E(i)$ or $E(j)$. Thus, for example, two edges in $P$, $(i, j)$ and $(i', j)$, may charge their weight to the same edge $(i, i')$ in the optimal solution. Clearly, this "double" charging can happen at most twice for each edge in the optimal solution, hence:

$$W(P) \leq 2W^*.$$

■

In the following, we show that our analysis of the upper bound of 2 on the approximation factor (proved in the lemma) is asymptotically tight. Consider the example in Fig. A.1c. Let us assume we want to zero out one element in each row and column in the block of size $4 \times 4$ presented in Fig. A.1a using the 2-approximate algorithm (Algorithm 1). First, we need to convert the block into a bipartite graph (as suggested in Figure 4). This construction appears in Fig. A.1b. Next, we sort the the edges from light to heavy and go over the sorted list. In Fig. A.1c we show the seven iterations of the 2-approximate algorithm. All edges are added to the list of chosen edges $P$ up until the 7th iteration. The algorithm stops at the 7th iteration, since after adding edge $u_4 \xrightarrow{1} v_4$, every node is already "covered" by at least one edge (in other words, each row and each column has at least one entry chosen for pruning). Note that the optimal solution would choose the edges that correspond to the entries on the diagonal (i.e., $u_1 \xrightarrow{1} v_1, u_2 \xrightarrow{1} v_2, u_3 \xrightarrow{1} v_3$, and $u_4 \xrightarrow{1} v_4$), summing up to a total weight of 4. Hence, we get an approximation ratio of $\frac{7}{4}$. It is easy to see that when using the same construction for a general block of size $M \times M$, we get an approximation ratio of $\frac{2M-1}{M}$, asymptotically converging to 2 as $M$ goes to infinity.

## A.3  Min cost flow vs. 2-approximation run-time analysis

In this section we specify the complexity of different min-cost flow methods and compare them with the running time of our 2-approximation method. Ahuja et al. [1] specifies the running times of six min-cost flow algorithms, two of which have distinctively better performance for our construction compared to the others (see Ahuja et al. [1][pp. 396-397]). The running time of these two methods depends on the following parameters: number of nodes $n$, number of edges $m$, the largest weight coefficient $W$, and the flow demand $U$. The Goldberg-Tarjan algorithm and the double scaling algorithm have running times of $\tilde{O}(mn)$, where $\tilde{O}$ hides polylogarithmic factors. Thus, in our construction, for a block of size $M \times M$ the number of edges is $M^2 + 2M$, the number of nodes is $2M + 2$, and the flow demand is $0.5M^2$. This boils down to running times of $\tilde{O}(M^3)$ for finding an optimal mask.The running time of the 2-approximation algorithm in comparison is $\tilde{O}(M^2)$. In Table A.3 we show the running time overhead of ResNet50 training using 2:4 transposable mask with IP, min cost flow and 2-approximation algorithms over regular training. These overheads are smaller than what we measured for 4:8 (Table 2). These methods were implemented in a non-optimized way, therefore, we expect a further decrease of the overhead in the 2-approximation in an optimized algorithm.

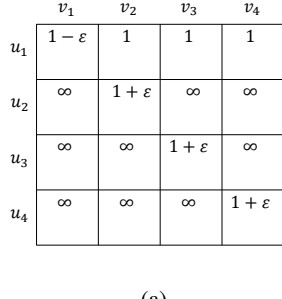 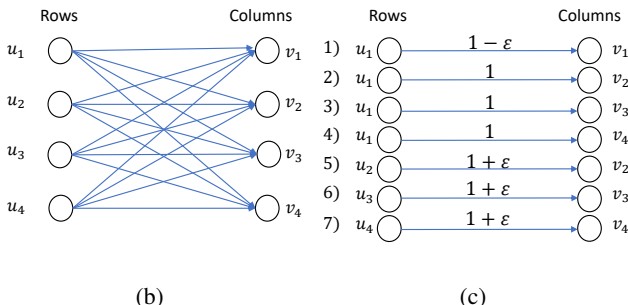

|  | (a) | (b) | (c) |
|---|---|---|---|

Figure A.1: **(a):** Block of size $4 \times 4$ where we want to zero out one element in each row and column using the 2-approximation algorithm. **(b):** The block represented as a directed bipartite graph. **(c):** The 7 iterations of the 2-approximation algorithm on the bipartite graph. Notice that we get an approximation ratio of $\frac{7}{4}$, since the optimal solution picks only the diagonal entries.

Table A.3: Overhead of different algorithms for finding the 2:4 transposable mask: ratio of their running time over regular training (ResNet50). Notice the overhead reduction in the 2-approximation algorithm in comparison to the naive IP.

| Method | Overhead (%) |
|---|---|
| Integer-programming | 160 |
| Min-cost flow | 65 |
| 2-approximation | 13 |

## A.4 Experiments Setting

In all our experiments we use 8 GPU GeForce RTX 2080 Ti.

**AdaPrune**    We used a small calibration set of 1000 images (one per-class). We run AdaPrune for 1000 iterations with batch-size of 100. For the results in the supplementary material, we kept the first and last layers dense.

**N:M transposable sparsity mask from a pre-trained model**    We used torchvison [28] model-zoo as our pre-trained dense baseline. For all ResNet models we used the original regime as given by He et al. [16], i.e., SGD over 90 epochs starting with learning rate of 0.1 and decreasing it at epochs 30,60,80 by a factor of 10. For BERT-large and MaskRCNN we used the defaults scripts as in Nvidia [35].

**N:M transposable sparsity mask from scratch**    We use the exact same setting as given by Zhou et al. [43].

## A.5 N:M hardware requirements

For conventional hardware, Broadly, N:M sparsity requires adding an N+1 to 1 multiplexer (N+1:1)to the adder tree in the code of the matrix multiplication. engine. Thus switching from 2:4 fine grained sparsity to 4:8 requires 5:1 multiplexers instead of 3:1. The simplest implementation of a multiplexer is build of set of 2:1 multiplexers which means that the area required for the multiplexers scale logarithmicly with the number of zeros in the block (N).

## A.6 Mask diversity Derivation

Let us consider $W$ to be a block of size $n \times n$ from a weight tensor and our desired sparsity level to be $N/M$. Thus, the MD of unstructured sparsity consists of all possibilities to pick $N$ values out of $B$ (Eq. (2) (a)). The MD increases with the block size ($B$), which might explain the recent

success of global pruning. Next we investigate, fine-grained $N : M$ structured sparsity [36]. This approach requires us to zero out $N$ values in each block of size $M$. Since we have $\frac{T}{M}$ blocks this results in Eq. (2) (b). If we wish to enforce the constraints on both row and columns of the matrix (i.e., $N : M$ transposable structured) the diversity decreases. Let us first assume $N = 1$. The number of possibilities in each block of size $M^2$ is $M!$. Repeating this process for general $N$ in all the $\frac{B}{M^2}$ blocks results in Eq. (2) (c). A more constrained mask, is a fine-grained $N : M$ mask with a sequential structure. Here we require that each $M$ contiguous elements would contain $N$ sequential zeros. In each block of size $M^2$, there are $M - N + 1$ options of sequential zeros. Applying it on all the $\frac{B}{M}$ blocks results in Eq. (2)(d).

### A.7 Mask diversity Experiments

In additional to the results in Section 3 we experimented in Fig. A.2 with ResNet50 over ImageNet dataset. In all our experiments we used one-shot pruning from dense model and applied the same regime as the original training as suggested by Nvidia [36].

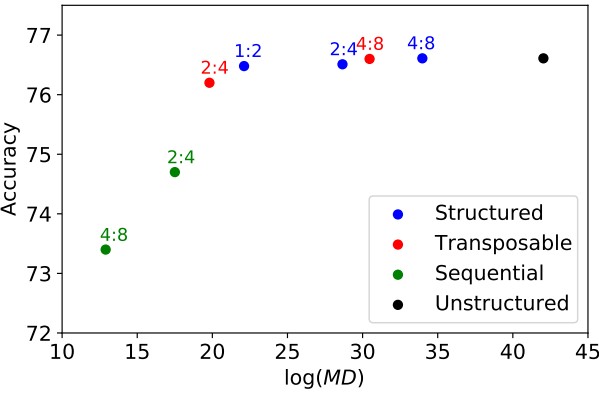

Figure A.2

Figure A.3: ResNet50 over ImageNet top-1 accuracy for weight sparsity of 50 % using different structured and unstructured masks. As expected the mask diversity correlates with the pruned model accuracy.

### A.8 2:4 transposable mask

In Table A.4 we show experiments with the 2:4 transposable mask in the "training from scratch" setting. Moreover, after we published the first version of our paper, researchers from NVIDIA [39] continued our work and demonstrated on a large set of models that one can achieve less than 1% degradation even with 2:4 transposable masks in the "training from a trained dense model" setting. As can be seen, 2:4 transpose mask can achieve high accuracy in part of the models. This results correlates with the shown MD, since 2:4 transpose mask has similar MD to 1:2 mask which already achieved high accuracy (Fig. A.2). Despite this, in some scenarios 2:4 transposable does not work as well, as in the case of finetuning BERT-Large on SQuAD dataset. Here the 2:4 transposable mask incurred a $\sim 1\%$ degradation (90.18 F1 vs. 91.1 F1 for the dense model) while a 4:8 transposable mask incurred less than 0.5% degradation in F1 score (90.65 F1).

Table A.4: Training from scratch of ResNet18, ResNet50 on imageNet dataset and fine-tuning of Bert-Large on SQuAD dataset with 2:4 transposable mask using the proposed 2-approximation scheme. SU refers to the sparse tensor cores utilization, used both in forward and backward passes in the proposed method, allowing 2x speedup in comparison to previous methods [43]

| Model | Metric | Baseline | | Ours 2:4-T | |
|---|---|---|---|---|---|
| | | SU | Accuracy | SU | Accuracy |
| ResNet18 | Top1 | | 70.54% | | 70.5% |
| ResNet50 | Top1 | 0% | 77.3% | 66% | 77.1% |
| Bert-Large | F1 | | 91.1% | | 90.18% |