# OpenReview forum: "Accelerated Sparse Neural Training: A Provable and Efficient Method to Find N:M Transposable Masks"
_NeurIPS.cc/2021/Conference — NeurIPS 2021 Poster_

### Official Review · Reviewer_o2wE · 2021-07-07

**Rating:** 6
**Confidence:** 5

**Summary:**

The sparse tensor core introduces the N:M sparsity in the latest Nvidia Ampere GPUs,  which bring sparse neural network deployment into the mainstream.  Most methods only focus on improving sparse model performance and compression ratio rather than speed up the training phase.



**Limitations And Societal Impact:**

The main flaw of this paper may be the gap between theory and practical.  I look forward to authors give more discussion.

**Main Review:**

This paper proposes a method to utilize ampere GPUs sufficiently by exploring N:M sparse training. To find the optimal transposable mask, this paper formulates the problem as an integer program solved by an approximated algorithm (2-Approximationalgorithm).

Strength:
      1. Direct sparse training using modern hardware is an important direction to train gigantic models [2].
      2. The 2-Approximation algorithm is novel and efficient.
      3. The performance of ResNets and Bert models is persuasive.
      4. The code is available and can reproduce the results.

Typo: line 273  "N   :  M "

The main flaw of this paper may be the gap between theory and practical.  I look forward to authors give more discussion.

Overall, this paper goes out an important step about fully sparse training, which should draw more attention in the future.

I am inclined to accept this paper.

**Time Spent Reviewing:**

7

---

> ### Author Response · Authors · 2021-08-10
> **Answer to Reviewer o2wE**
>
> We would like to thank the reviewer for his encouraging feedback. Please see our answers below.
>
> $\textbf{A1}$ Our paper proposed the use of an N:M transposable fine-grained mask to accelerate training (Forward and backward pass) in comparison to previous methods (ASP [34] and N:M-SS [40])  which were able only to accelerate the forward pass. The paper suggests a new measure called mask-diversity (MD) to compare different sparsity patterns. We showed MD correlates with the network accuracy and therefore suggest to use 4:8 transposable mask which has similar MD to the previously used 2:4 mask. We showed in Tables 3 and 4 that we achieved with 4:8 transposable mask similar results to 2:4 mask while increase by 2x the sparse tensor core utilization. We agree with the reviewer that the current ampere GPU only supports 2:4 pattern and this can be a gap between the theory and practical use. Therefore we experimented with 2:4 transposable mask in appendix A.7 well. As can be seen in figure A.2, 2:4 transposable mask results in a minor accuracy degradation. We discussed in detail our 2:4 transposable mask experiment in the general comment above ("2:4 works almost as well as 4:8").
>
> ------------------------------------------------------------------------------------------------------------------------------------------------------------------------------------
> [34] Nvidia. a100 tensor core gpu architecture. 2020. URL https://www.nvidia.com/content/dam/en-zz/Solutions/Data-Center/nvidia-ampere-architecture-whitepaper.pdf
>
> [40] Aojun Zhou, Yukun Ma, Junnan Zhu, Jianbo Liu, Zhijie Zhang, Kun Yuan, Wenxiu Sun, and Hongsheng Li. Learning n:m fine-grained structures sparse neural networks from scratch. In ICLR, 2021

---

> ### Comment · Reviewer_o2wE · 2021-08-18
> **My concern has been addressed**
>
> Thanks for the detailed response. I am very much looking forward to these changes in the paper.

---

### Official Review · Reviewer_E3cM · 2021-07-13

**Rating:** 8
**Confidence:** 5

**Summary:**

The authors study N:M structured sparsity, specifically the challenge of finding N:M transposable masks so that N:M sparsity hardware acceleration can be exploited for part of the backward pass computation in addition to the forward pass. The authors additionally tackle the challenge of adapting models trained with other kinds of unstructured or structured sparsity to specific topology constraints supported by hardware. The authors provide a detailed analysis of the problem of finding transposable N:M masks and asymptotics associated with solving the problems and present a practical solution that achieves good results in practice.

**Ethical Concerns:**

I did not identify ethical issues with this paper.

**Limitations And Societal Impact:**

I did not identify potential negative societal impact in this work.

**Main Review:**

Overall the paper is very well done. There are a few issues in the experiments I’d like to see resolved and then I believe this work will be an excellent contribution to the community.

Introduction and Background
1. Structure causing accuracy degradation in sparsification is not a hypothesis. It’s well documented empirically [1, 2].
2. “However [unstructured sparsity’s] hardware cost makes it infeasible to use on modern architectures“. Recent work has shown that unstructured sparsity can be accelerated, although it still requires high sparsity to exceed the performance of dense computation [1, 3].

Experiments (Changes Required)
1. Table 3, claiming a ‘2x” speedup over ASP is not correct. 2x more of the FLOPs in the model can now be accelerated by sparse tensor cores, which provide up to 2x performance over dense tensor cores. Let's say dense models speed is (fwd + bwd + update) / dense_throughput. For simplicity, we’ll say fwd = bwd = update so baseline takes 3 * fwd / dense_throughput seconds. The ASP model would then take fwd / (2 * dense_throughput + 2 * fwd / dense_throughput = 2.5 * fwd / dense_throughput. Your model would then take 2 * fwd / (2 * dense_throughput) + fwd / dense_throughput = 2 * fwd / dense_throughput. Thus, ASP provides a 3 / 2.5 = 1.2x speedup over the baseline and you would provide a 2.5 / 2 = 1.25x speedup over ASP. In reality your speedup will be lower because not all operations (e.g., batch normalization, activation functions) are accelerated by sparse tensor cores and the sparse tensor core operations will not fully realize the theoretical 2x gain over the dense tensor cores. This should be corrected in the authors’ response.
2. I understand that 4:8 transposable masks have a similar MD to 2:4 masks but comparing 4:8 sparsity to 2:4 sparsity is not apples-to-apples as I expect 4:8 sparsity will make it much easier to maintain model quality. I like the inclusion of 4:8 sparsity results, but I’d also like to see 2:4 transposable results to see what the accuracy loss looks like in comparison to non-transposable masks.
3. In Table 2, what N:M are used? As I understand it, the algorithm complexity depends on these factors.
4. X-axis in Figure 5 (b) should be scaled by 100 to be percent.

Experiments (Changes Suggested)
1. The hypothesis that MD is correlated with accuracy would be made much stronger with results on other kinds of structured sparsity. For example, block sparsity and channel-sparsity. As someone who works on this topic, I’d be very interested to see these results.
2. In Table 1, it might be nice to include the MD normalized to unstructured as well. This would make it easier to compare across the different structures for the reader.
3. Most sparse training algorithms only perform mask updates periodically to reduce overhead (and also because it usually leads to higher accuracy). The update period is usually ~100 (e.g., with RigL). 40 is a tighter bound, but you could also run 100 (in Table 2) to align with prior work.

Organizational, Spelling and Grammar
1. On page 2, your 3 questions are in a different order than they’re shown in your diagram (Figure 4)
2. “sparse core tensor” should be “sparse tensor core”
3. “|” before “In” should be deleted.

Additional Improvements
1. Nvidia has recently proposed using permutations to reduce accuracy loss (talk, poster from SNN). How do permutations affect your MD metric? Do the results still correlate with accuracy? Do they make finding transposable N:M masks more difficult? These might be interesting questions to discuss in your conclusion.

References
1. https://arxiv.org/abs/1911.09723
2. https://arxiv.org/abs/1811.00206
3. https://arxiv.org/abs/2006.10901


**Time Spent Reviewing:**

4hrs

---

> ### Author Response · Authors · 2021-08-10
> **Answer to Reviewer E3cM**
>
> We would like to thank the reviewer for his detailed review and positive feedback. Please see our answers below.
>
>
> $\underline{\textbf{Introduction \hspace{2pt} and \hspace{2pt}  Background}}:$
>
> $\textbf{A1}$ Thank you for these references which support our work. We'll fix the final version according to your comments and add these references.
>
> $\underline{\textbf{Experiments \hspace{2pt} (Changes \hspace{2pt} Required)}}:$
>
> $\textbf{A2}$  We agree with the reviewer that the '2x' speedup statement isn't correct and that the maximum speedup is 1.25x. As written in Tables 3 and 4 the correct term is '2x' sparse tensor core utilization (Fwd + Bwd in our method in comparison to only Fwd in ASP). We'll fix it in the revised manuscript.
>
> $\textbf{A3}$ In section A.7 of the appendix (figure A.2) we plotted the accuracy for R50 with 2:4 transposable mask for the ASP setting. Following your comment, we perform a more thorough analysis of 2:4 transposable mask and discuss it in detail in the general comment above ("2:4 works almost as well as 4:8"). We intend to add those results to the revised manuscript.
>
> $\textbf{A4}$ In Table 2 we used 4:8 transposed mask, which was the main focus of our work. As mentioned by the reviewer, the algorithm should depend on N:M, however as written in lines 223-224 they were implemented in a non-optimized way. Generally, as the block size increase the overhead increase and thus we expect a reduced overhead for the 2:4 experiments.
>
> $\textbf{A5}$ Thank you for the thorough review, we'll fix the graph axis in the revised manuscript.
>
> $\underline{\textbf{Experiments \hspace{2pt} (Changes \hspace{2pt} Suggested)}}:$
>
> $\textbf{A6}$ We showed in Table 1 the MD and his correlation to accuracy for "sequential N:M sparsity", i.e any block of size M must contain N sequential zeros - which is a type of block-sparsity pattern. With respect to channel pruning, we agree with the reviewer that it will be interesting to see there the correlation of MD with accuracy - we plan to add such experiments in the next version of the manuscript.
>
> $\textbf{A7}$ We agree with the reviewer that it will be more clear for the reader to normalize the results in Table 1. We'll fix it in the next version.
>
> $\textbf{A8}$ We experimented ResNet18 with 4:8 transposable mask, increasing the period between updates to 100 iterations and get a similar result (70.4\%).
>
> $\underline{\textbf{Organizational, Spelling \hspace{2pt} and   \hspace{2pt} Grammar}}:$
>
> $\textbf{A9}$ thank you for the thorough review, we'll fix all the grammar errors mentioned by the reviewer in the next version.
>
> $\underline{\textbf{Additional \hspace{2pt} Improvements}}:$
>
> $\textbf{A10}$ We believe the reviewer refers to the work "Channel permutations for N:M sparsity" presented in SNN workshop 2021. This work suggests a pre-processing method for permuting the channels in order to increase the magnitude of the masked weight for the corresponding N:M mask. Since this method is a pre-processing method for the weights, it shouldn't affect the MD which refers to the number of combinations after any pre-processing method. Therefore we argue that under channel permutation (CP), MD should have a similar correlation with model accuracy. Yet we expect channel permutation to increase the sparse model accuracy for all sparsity patterns. Since CP requires extracting the mask for each considerable permutation applying CP to any mask-finding algorithm would increase the running time. While this work was presented after we submitted our work, it would be interesting to find out if it can shorten the number of iterations required to regain accuracy. We'll add such discussion in the conclusions of the final version.

---

> > ### Comment · Reviewer_E3cM · 2021-08-13
> > **Reviewer Response**
> >
> > Thank you for your reply. With the addition of 2:4 sparsity results and the changes I requested I think the paper is greatly improved. I've increased my score based on these additions. This work is well done and valuable for the community - I think it should be accepted.

---

### Official Review · Reviewer_PZCA · 2021-07-16

**Rating:** 6
**Confidence:** 4

**Summary:**

The paper proposes a method to quickly find masks that allow DNN models to train with N:M sparsity. Prior works mainly leverage N:M sparsity for inference (i.e., forward pass), but training cannot directly benefit from N:M sparsity because model weights are transposed during backpropagation to calculate gradients. Therefore, the N:M sparsity constraints may no longer be satisfied during backward pass. The paper suggests that it is possible to find so-called N:M transposable masks, which allow both the forward pass and the backward pass of a DNN training to perform N:M sparse computation. One key challenge in this task is to find such transposable masks in a lightweight way since sparsity masks may need to dynamically change in order to get on-par accuracy as dense training. To quickly find masks that satisfy the N:M transposable mask constraints, the paper proposes to use a greedy-based approximation method. The authors show that such a method adds a modest amount of runtime overhead and scales almost linearly with respect to the input sizes. The paper conducts evaluations against some computer vision (e.g., ResNet) and NLP tasks (e.g., BERT fine-tuning tasks) to show that such a training regime can bring comparable accuracy as dense training.


**Limitations And Societal Impact:**

Please see the limitations in the detailed comments.

**Main Review:**

Strengths:

-- The transposable mask is an interesting idea to enable the system capability of N:M sparse training.
-- The proposed 2-approximation solution adds a low overhead in finding transposable masks.

Weaknesses:

-- The proposed transposable N:M masks add strong constraints to weight diversity, which may limit the expressiveness of the model.
-- No real training speedups from the proposed method, because real hardware constraints are not satisfied.
-- Lack of details of implementation and hyperparameter settings.
-- Baseline results are inconsistent.
-- No discussions on how model convergence or training dynamics are affected by N:M sparse training.

Comments:

1. The transposable N:M sparsity masks seem to add very strong constraints to the weight matrices, which might limit the model's expressiveness in a significant way. Indeed, after applying transposable N:M sparsity masks, there won't be enough mask diversity for 2:4 sparsity. As a result, the paper has to find the transposable N:M sparsity in 4:8. This is a deviation from real hardware constraints because no existing hardware supports 4:8 sparsity (Nvidia sparse tensor core supports 2:4).  The proposed method, therefore, cannot speed up training on real hardware either. It is disappointing as the paper motivates itself as a way to accelerate sparse training on real hardware in the introduction and only to find out later that the solution does not really work on real hardware.

2. Some of the results of the evaluations are inconsistent, especially the baseline results. While the baseline results for ResNet18 and ResNet50 are 69.7% and 76.1% in Table 3, they become 70.5% and 77.3% in Table 4. Why are there two results for the same dataset and task? Which baseline should we look at? If we take 77.3% as the baseline result for ResNet50, then the proposed N:M sparse training actually leads to a non-trivial drop (77.3 vs. 76.6) in Table 3, and the "no accuracy degradation" claim is also not true. Please double-check the results and also provide detailed hyperparameter information.

3. Despite claiming 2x speedup, no real training time is reported for all evaluations. The training cost depends on both the training time per iteration as well as the total number of iterations needed to converge. Although N:M sparse training might reduce the training time per iteration by 2x,  if the N:M sparse training takes more iterations to converge to get the same accuracy or introduces training instability, then N:M sparse training cannot really reduce the training cost and reducing the cost of finding transposable N:M sparsity becomes even less motivated.  Furthermore, the 2x speedup claim also seems to be problematic because (1) finding the transposable N:M masks still add non-negligible cost, which would make the speedup to be less than 2X, and (2) even with sparse tensor core, one cannot expect a 2x speedup because a DNN contains more components than just the matrix multiplication, such as non-linear functions or BatchNorm, which cannot be accelerated by sparse tensor core.


4. The paper lacks details for implementation and hyperparameter settings. Can the authors add details for how the model has been trained, such as the number of update steps, the batch size, etc. for both the baseline and the N:M sparse training, as well as parameters for AdaPrune? The current paper doesn't seem like it could be reproduced.

5. The proposed AdaPrune only seems to convert models trained with 8:4 to 4:2 for inference. Can it also be used to accelerate the training speed as well?

6. The writing of Section 5 seems to be not good, with many typos and grammar errors. Just to give a few examples:

In line 223, "…over regular training, The algorithms…" -> Change "," to ".".
In line 249, "…where ASP do not." -> "does not"
In line 261, "…to accelerate training. but what should we do…" -> Change "." to ",".
In line 262, "…wish to deployed the output…" -> "wish to deploy".

-------------------------

After a series of discussions, the authors have addressed my concerns. I have increased my score from 4 to 6.


**Time Spent Reviewing:**

6 hours

---

> ### Author Response · Authors · 2021-08-10
> **Answer to Reviewer PZCA**
>
> We thank the reviewer for his comments, please see our answers below:
>
> $\textbf{A1}$ The N:M transposable mask and the methods to generate them are defined in a generic fashion. The different structural mask constraints are then discussed in Section 5 (e.g., Figure 6). There, we show that transposable masks can be applied in various sparsity patterns with only minor degradations, including the 2:4 sparsity pattern. For example, see Figure A.2 for the 2:4 transposable mask in the ASP setting (i.e., "initialized from a trained dense model"). We discuss more thoroughly our 2:4 mask results in the general comment above (" 2:4 works almost as well as 4:8").
>
> $\textbf{A2} \hspace{2pt}$  In section 4.1 we discuss two independent types of experiments for the N:M transposable mask. In each case, we compare with relevant results from previous work and their own baseline. Specifically, for the "initialized from a trained dense model" experiments we use the torchvision model-zoo as the baseline (as it has a large set of pretrained models) and compare with ASP [34].  For a fair comparison, in the "training from scratch" experiments we use the N:M-SS [40]  settings and run the baseline models. We further discuss implementation details in appendix A.4.
>
> $\textbf{A3}$ As stated in section A.4 of the appendix, in all the reported experiments we used the \textit{"original regime"} (line 573) and \textit{"exact same setting"} (line 577). That means also the same number of iterations. Additionally, for the "training from scratch" experiment we showed (Table 2) that the overhead for finding the N:M transposable mask using our \textbf{non-optimized} implementation is only 14\% for a single iteration. Since we change the mask once in 40 iterations (line 224) the overhead is insignificant. In the "initialized from a trained dense model" setting the overhead is even more negligible since we find the mask only once before training begins.
>
> Regardless, we agree with the reviewer that the acceleration refers only to the matrix multiplication operations (done by the tensor cores in NVIDIA's GPUs) and not to the scalar operations (done by Cuda cores in NVIDIA's GPUs). We will clarify it in the final manuscript. However, we note that matrix multiplication tends to be the bottleneck in modern accelerators. As evidence, NVIDIA increased the tensor-core to Cuda-cores throughput ratio in its latest architecture (Amper). V100 has 125 TFLOPS tensor cores and 15.7 TFLOPS of Cuda cores. A100 has 312 TFLOPS of tensor cores and 19.5 TFLOPS of Cuda cores. This emphasizes the importance of this work - accelerating the matrix multiplication engine.
>
> $\textbf{A4}$ Perhaps the reviewer missed the last line in our abstract, where we give the full reference implementation, with scripts to reproduce the results in the paper. Additionally, in appendix A.4 we state the setting for all our experiments, "\textit{for the "N:M initialized from a trained dense model" experiments we use torchvision model-zoo as the baseline and trained with SGD over 90 epochs, batch size 256, with starting l.r of 0.1 and decreasing at epochs 30,60,80 by a factor of 10. For the "Sparse training from scratch" experiments we use the same hyperparameters as Zhou et al [40]. i.e SGD optimizer, batch size 256, l.r 0.1 with cosine annealing scheduler. For the AdaPrune experiments, we use 1000 images as the calibration set, run them for 1000 iterations with batch size 100.}
>
> $\textbf{A5}$ AdaPrune is a very light method for converting between models sparsity patterns for inference (e.g., 2:4 to 1:2 see Appendix A.1). Generally AdaPrune cannot be used to accelerate training, yet we show in appendix A.1 it can be used for convert a dense pre-trained model to an N:M sparse model.
>
> $\textbf{A6}$ Thank you for pointing out these grammars errors. We'll fix them in the revised manuscript.
>
> -------------------------------------------------------------------------------------------------------------------------------------------------------------------------------
> [34] Nvidia. a100 tensor core gpu architecture. 2020. URL https://www.nvidia.com/content/dam/en-zz/Solutions/Data-Center/nvidia-ampere-architecture-whitepaper.pdf
>
> $[40]$ Aojun Zhou, Yukun Ma, Junnan Zhu, Jianbo Liu, Zhijie Zhang, Kun Yuan, Wenxiu Sun, and Hongsheng Li. Learning n:m fine-grained structures sparse neural networks from scratch. In ICLR, 2021

---

> > ### Comment · Reviewer_PZCA · 2021-08-23
> > **Comments to authors' responses**
> >
> > The reviewer appreciates the authors' response. The updated results of the 2:4 sparse training look promising but seem to cover only the accuracy impact. If I understand correctly, the main technical contribution of the paper is on proposing an approximation method to find the N:M transposable masks, and Table 2 reports the overhead of finding 4:8 transposable masks. Therefore, a lot of the performance results related to 2:4 sparsity are not reported in the main paper. What is the overhead of finding 2:4 transposable masks, using IP, min-cost flow, and 2-approximation? Also, what is the fraction of 2:4 transposable masks that get changed in between iterations when training from scratch?

---

> > > ### Author Response · Authors · 2021-08-24
> > > **Answer to Reviewer PZCA  (2)**
> > >
> > > We thank the reviewer for the response, please see our answers below:
> > >
> > > $\textbf{Q1)}$  "If I understand correctly, the main technical contribution of the paper is on proposing an approximation method to find the N:M transposable masks, and Table 2 reports the overhead of finding 4:8 transposable masks."
> > >
> > > $\textbf{A1)}$ Just to be on the same page, we would like to clarify that our paper has several additional contributions. Except for the approximation method to find the N:M transposable masks, we are also the first work which suggests the use of such transposable masks. Additionally, we use a measure (“mask diversity”) which connects between mask constraints and network accuracy. Finally, we suggested a light method called AdaPrune that can change the type of sparsity mask without the need of full re-training.
> > >
> > > $\textbf{Q2)}$ "a lot of the performance results related to 2:4 sparsity are not reported in the main paper. What is the overhead of finding 2:4 transposable masks, using IP, min-cost flow, and 2-approximation?"
> > >
> > > $\textbf{A2)}$  We run the same experiment as in Table 2 for 2:4 transposable mask and get the following overheads:
> > >
> > > IP - 160%
> > >
> > > Min cost flow - 65%
> > >
> > >  2-approximation - 13%
> > >
> > > Unsurprisingly, these overheads are all somewhat smaller than what we measured for 4:8 (which was 180%,70%, and 14% respectively). Moreover, as written in lines 223-224, these methods were implemented in a non-optimized way. Therefore, we expect a further decrease of the overhead in the 2-approximation in an optimized algorithm.
> > >
> > > $\textbf{Q3)}$ "Also, what is the fraction of 2:4 transposable masks that get changed in between iterations when training from scratch?"
> > >
> > > $\textbf{A3)}$ The “training from scratch” experiments with 2:4 transposable masks follow the exact same settings as the others “training from scratch” experiments, i.e. we update the mask every 40 iterations.  Additionally, following the request of reviewer E3cM we did one experiment where we updated the mask every 100 iterations (ResNet18) and got a similar accuracy (70.4%).

---

> > > > ### Comment · Reviewer_PZCA · 2021-08-24
> > > > **Reviewer response**
> > > >
> > > > Thank you for the reply. It would be better to add the 2:4 performance results in the paper.
> > > >
> > > > Regarding Q3, my question is actually about the dynamism of the masks. Think about two iterations A and B, at iteration A the 2:4 mask found is M_A, and at iteration B, the mask found is M_B. How many mask positions get changed in between A and B? Is it possible that once the masks have been decided once their positions stay the same in the entire training process?

---

> > > > > ### Author Response · Authors · 2021-08-24
> > > > > **Answer to Reviewer PZCA (3)**
> > > > >
> > > > > $\textbf{Q1)}$ It would be better to add the 2:4 performance results in the paper.
> > > > >
> > > > > $\textbf{A1)}$ Of course, we will add these results in the final version of the paper.
> > > > >
> > > > > $\textbf{Q2)}$ Regarding Q3, my question is actually about the dynamism of the masks. Think about two iterations A and B, at iteration A the 2:4 mask found is M_A, and at iteration B, the mask found is M_B. How many mask positions get changed in between A and B? Is it possible that once the masks have been decided once their positions stay the same in the entire training process?
> > > > >
> > > > > $\textbf{A2)}$ In the first part of the training we noticed that the mask completely changes in every update. However, we see less and less changes with the progress of the training and the reduction of the learning-rate. We agree with the reviewer this is an interesting ablation study and we will add it in the final version of the paper.
> > > > > Notice that doing "training from scratch" without changing the mask at all leads to very poor results, which is the main motivation for [40]  using N:M mask. We saw the same behavior for N:M transposable mask.
> > > > >
> > > > > ------------------------------------------------------------------------------------------------------------------------------------------------------------
> > > > > $[40]$ Aojun Zhou, Yukun Ma, Junnan Zhu, Jianbo Liu, Zhijie Zhang, Kun Yuan, Wenxiu Sun, and Hongsheng Li. Learning n:m fine-grained structures sparse neural networks from scratch. In ICLR, 2021

---

### Author Response · Authors · 2021-08-10
**General comment: 2:4 works almost as well as 4:8**

All reviewers wondered why our method relies on 4:8 block sparsity instead of the common 2:4 block sparsity available in current hardware. Thus in the following, we: (1) demonstrate that our transposable method works well also with 2:4 sparsity for most vision models; (2) explain how this is predicted by the Mask diversity measure; and (3) show that 4:8 sparsity is useful in more constrained models, and is, therefore, a potentially interesting direction for new hardware.

$\textrm{(1) Results for 2:4 sparsity.}$ In appendix section A.7 (figure A.2) we plotted the accuracy for R50 with 2:4 transposable mask for the ASP [A] setting (i.e., "initialized from a trained dense model"). As can be seen, we incur a small accuracy degradation 76.16\% top-1 accuracy vs. 76.6\% with 4:8 transposable mask.

Moreover, after we published the first version of our paper, researchers from NVIDIA [B] continued our work and demonstrated on a large set of models that one can achieve less than 1\% degradation even with 2:4 transposable masks (named "2:4 2D" in [B]. See Table 2, "2:4 2D" columns).

Additionally, our latest experiments show that the 2:4 transposable mask can be applied effectively also in the "training from scratch" setting: ResNet18 achieved 70.5\% top-1, ResNet50 achieved 77.1\% top-1, similar to 4:8 transposable.

$\textrm{(2) Why does 2:4 sparsity mask often work well?}$  Our starting point was the ASP paper [A] which focused on 2:4 mask and demonstrated on par results for many models. To obtain a similar accuracy with a transposable mask, our MD measure suggested we should use a 4:8 transposable mask. However, the surprising high accuracy of 2:4 transposable should have been obvious in retrospect from Figure A.2: there, the 2:4 transposable mask has a similar MD as the 1:2 mask, and the 1:2 mask also has high accuracy (76.5\% top-1 accuracy). In other words, the ResNet50 can handle more rigid sparsity constraints than 2:4 mask. We further experimented with the 1:2 mask and noticed that it can be used for many vision models without harming accuracy (for instance ResNext50 77.5\% top-1 using ASP's setting).

$\textrm{(3) Relevance of 4:8 sparsity.}$ Despite the above results, in some scenarios 2:4 transposable does not work as well, as in the case of finetuning BERT-Large on SQuAD1.1 dataset. Here the 2:4 transposable mask incurred a 1\% degradation (90.18 F1 vs. 91.1 F1 for the dense model) while a 4:8 transposable mask incurred less than 0.5\% degradation in F1 score (90.65 F1). In section A.5 we discuss the hardware requirements for 4:8 support and argue that 4:8 implementation is reasonable.

We plan to add those experiments to the final version of the manuscript since we believe they further stress the importance of our transposable masks and the MD measure.

[A] Nvidia. a100 tensor core gpu architecture. 2020. URL https://www.nvidia.com/content/dam/en-zz/Solutions/Data-Center/nvidia-ampere-architecture-whitepaper.pdf

[B] Stosic, Darko, and Dusan Stosic. "Search Spaces for Neural Model Training." arXiv preprint:2105.12920 (2021).

---

### Decision · Program_Chairs · 2021-09-27

**Decision:**

Accept (Poster)

**Comment:**

I want to thank the authors for their active engagement with the reviewers in the rebuttal period, and I'm happy this lead to some modification of the paper (as outlined by the authors) that I think greatly improved the work.

The proposed method seems novel and valuable to the community, coupled with thorough experimentation and a clear exposition, I think this paper matches the requirements for the conference.